# Changes in Sheep Behavior before Lambing

Beatrice E. Waters [1], John McDonagh [2], Georgios Tzimiropoulos [3], Kimberley R. Slinger [1], Zoë J. Huggett [1] and Matt J. Bell [4,*]

1   School of Biosciences, University of Nottingham, Sutton Bonington Campus, Sutton Bonington LE12 5RD, UK; Beatrice.Waters@nottingham.ac.uk (B.E.W.); kimberley.slinger@nottingham.ac.uk (K.R.S.); Zoe.Huggett3@nottingham.ac.uk (Z.J.H.)
2   School of Computer Science, University of Nottingham, Jubilee Campus, Nottingham NG8 1BB, UK; john.mcdonagh@nottingham.ac.uk
3   School of Electrical Engineering and Computer Science, Queen Mary University of London, London E1 4NS, UK; g.tzimiropoulos@qmul.ac.uk
4   Agriculture Department, Hartpury University, Gloucester GL19 3BE, UK
*   Correspondence: matt.bell@hartpury.ac.uk

**Abstract:** The aim of this study was to assess the duration and frequency of behavioral observations of pregnant ewes as they approached lambing. An understanding of behavioral changes before birth may provide opportunities for enhanced visual monitoring at this critical stage in the animal's life. Behavioral observations for 17 ewes in late pregnancy were recorded during two separate time periods, which were 4 to 6 weeks before lambing and before giving birth. It was normal farm procedure for the sheep to come indoors for 6 weeks of close monitoring before lambing. The behaviors of standing, lying, walking, shuffling and contraction behaviors were recorded for each animal during both time periods. Over both time periods, the ewes spent a large proportion of their time either lying (0.40) or standing (0.42), with a higher frequency of standing (0.40) and shuffling (0.28) bouts than other behaviors. In the time period before giving birth, the frequency of lying and contraction bouts increased and the standing and walking bouts decreased, with a higher frequency of walking bouts in ewes that had an assisted lambing. The monitoring of behavioral patterns, such as lying and contractions, could be used as an alert to the progress of parturition.

**Keywords:** behavior; birth; management; observations; sheep

## 1. Introduction

A stockperson's ability to assess animal behavior is a key component of their ability to recognize and treat ill-health, and evaluate the wellbeing of their livestock [1,2]. The visual assessment of livestock by humans is subjective and has several limitations such as the cost of labor and time to regularly observe individual animals. Hence, several monitoring technologies have been proposed in recent years that predict animal behaviors from movement sensors on cattle or sheep [3–8]. New technologies that provide an objective measure of animal behavior, such as sensors and cameras, could provide an aid to improve animal management [9]. Furthermore, monitoring equipment can continuously and remotely track livestock, something that would be unrealistic and too costly for human observers to replicate [3].

Lambing is a critical time in the productive life of sheep and the development of the newborn offspring that will eventually be sold or retained as flock replacements. Sheep will often have multiples at birth, which can be physically challenging, stressful and a painful process for the mother and offspring that may require a farmer's intervention [10]. The parturition period is associated with several physiological, hormonal and behavioral changes in the pregnant animal, with restless behavior exhibited by nesting and reduced appetite along with birth contractions, which increase in frequency and intensity as birth progresses [10,11]. Studying cows, Huzzy et al. [12] found a dramatic increase in the

number of positional changes such as lying or standing at calving, and reported that the animals tended to isolate themselves from the rest of the herd. Although there are several studies on changes in cattle behavior before calving [10,12–15], there are few studies on pregnant sheep [16,17]. The need for further research into enhanced monitoring approaches of sheep during parturition has been identified by others [17], and assist a stockperson during this critical period. The hypothesis of the current study was that there is a change in sheep behavior before giving birth, and this change can be visually observed. This insight may assist lambing management and future monitoring technologies.

The objective of this study was to assess the duration and frequency of behavioral observations of pregnant sheep as they approached lambing. The sheep studied followed normal husbandry procedure of being housed as a group at about 6 weeks before lambing to allow for closer monitoring.

## 2. Materials and Methods

Approval for this study was obtained from the University of Nottingham animal ethics committee before the study commenced (approval number 198, 2018).

### 2.1. Data

A total of 17 pregnant ewes were monitored using video camera surveillance (5 Mp, 30 m IR. Hikvision HD Bullet; Hangzhou, China) at the Nottingham University Farm (Sutton Bonington, Leicestershire, UK; 52.8282° N 1.2485° W, 48 m a.s.l) when indoors before lambing from February to March 2019. The study was designed to have similar numbers of primiparous and multiparous ewes. The ewes monitored were predominantly Lleyn breed, with 9 primiparous and 8 multiparous. Normal husbandry procedure for the flock were followed, whereby all sheep came indoors for closer monitoring as a single group at about 6 weeks before lambing, and returned to pasture after lambing. The sheep were group housed on straw bedding, with an open feed trough for forage and supplementary feed and a single water trough. A single camera was used to obtain continuous video footage of each ewe. The camera position allowed full coverage of the area and at an approximate 45-degree angle looking into the sheep pen. When the sheep were housed at the start of the study they were weighed, marked with a number for camera observations and individual identification recorded, and vaccinated for pasteurella and clostridial disease, but after this the sheep were not handled until they had given birth. The sheep did not receive any other health treatments, such for endo or ectoparasites, during the study. The average age of primiparous ewes was 1.9 (s.d. 0.03) years and multiparous 4.3 (s.d. 1.0) years. The average bodyweight of primiparous ewes was 57.9 (s.d. 2.6) kg and multiparous 64.2 (s.d. 6.4) kg. Sheep were group fed *ad libitum* haylage consisting of 9.7 MJ/kg for metabolizable energy (ME), 486, 548, 138, 59 g/kg for dry matter, neutral detergent fiber, crude protein and sugar, respectively (Sciantec Analytical Services, Cawood, UK; using near-infrared spectroscopy analysis). Additionally, sheep were supplemented with 350 g/day oats with wheat distillers grain mix (13.4 MJ/kg ME, 860, 250, 228 and 60 g/kg for dry matter, neutral detergent fiber, crude protein and sugar, respectively). The diet was about 75% forage on a dry matter basis. The feed was allocated as 2% of the average bodyweight for the group of ewes (about 1.2 kg/day), with the diet formulated based on an estimated energy and protein requirement of 11 MJ/kg ME and 160 g/kg crude protein in the diet [18]. The same diet was fed throughout the study and the amount allocated to the group was reduced as sheep gave birth and were removed from the lambing pen. Need for a birth to be assisted by farm staff was recorded for each ewe. The average daily temperature was 5.7 °C, rainfall was 2.9 mm and humidity was 90% during the study.

### 2.2. Observations

Two observation periods were used to investigate changes in behavior: Period 1 (4–6 weeks before lambing) and Period 2 (at lambing). There were 10 h of annotated video

recordings for each ewe from Period 1 and three hours before the first lamb was born for Period 2. Tracking of sheep in Period 2 before lambing was challenging because of the similar appearance and behavioral changes. Three observers used custom made scripts in PyTorch 1.5 framework to record the behavior profile of each ewe with time. A total of 8257 individual behavioral observations were recorded from all 17 ewes. To ensure accuracy of video behavior annotations, the video was segmented into short clips for each behavior, and all video clips subsequently checked for accuracy by one of the three observers. Five behaviors were recorded, which were:

1. Standing: The sheep is still on all four legs.
2. Lying: The midway transition of when the sheep is about to lie down to when they start to rise again.
3. Walking: Movement of more than two steps.
4. Shuffle: Sheep circles on the spot or moves slightly with a step or two.
5. Contractions: Visible straining while lying down.

*2.3. Statistical Analysis*

The duration of behaviors in seconds and behavior frequency were determined for both time periods. A total of 170 behavior records were obtained from 17 ewes (17 × 5 behaviors × 2 time periods).

Behavior records were analyzed using a generalized linear mixed model in Genstat Version 19.1 (Lawes Agricultural Trust, 2018). A binomial error distribution and a logit link function was fitted to the fixed effects of assistance, time period, behavior and parity for the dependent variables of duration and frequency of behaviors in Equation (1):

$$Y_{ijkl} = \mu + A_i \times T_j \times B_k + P_l + E_{ijkl} \tag{1}$$

where $Y_{ijkl}$ is the dependent variable of behavior duration or frequency; $\mu$ = overall mean; $A_i$ = fixed effect for assistance at lambing (i = 0 for unassisted or 1 for assisted); $T_j$ = fixed effect of time period (j = 1 or 2); $B_k$ = fixed effect of behavior (k = standing, lying, walking, shuffling, contractions); $P_l$ = fixed effect of parity (l = primiparous or multiparous); $E_{ijkl}$ = random error term.

The back-transformed predicted means for behavior duration and frequency were expressed as the proportion of total time or count during each time period. Significance was attributed at $p < 0.05$.

## 3. Results

Of the 17 lambings, two were triplets, 13 were twins and two were singles. There were four primiparous and two multiparous ewes that required assistance by the farm stockperson, with all other lambings being unassisted.

Differences were found in the duration of behaviors ($p < 0.001$) with most of the time spent either lying (0.40) or standing (0.42), with other behaviors being 0.08 or less across time periods studied (Figure 1).

There was no effect of parity, time period, lambing assistance on duration of behaviors ($p > 0.05$; Table 1 and Figure 2).

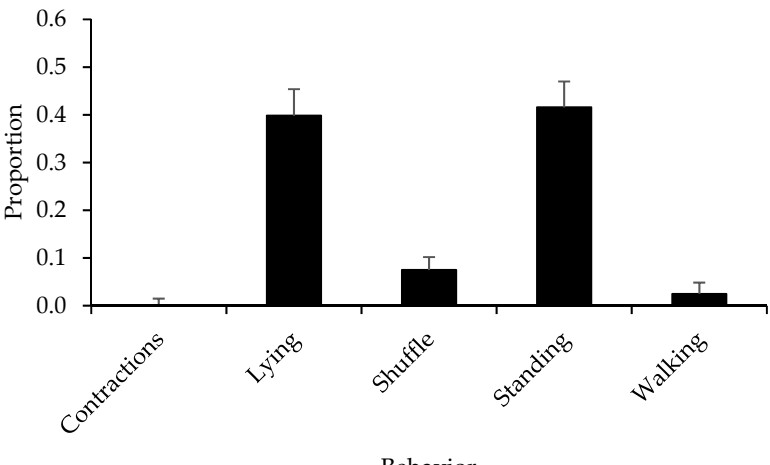

**Figure 1.** Predicted mean (±SEM) proportion of time spent in doing different behaviors.

**Table 1.** Effects of parity, time period, and lambing assistance on the duration of behaviors as a proportion of time.

| Variable | | Mean (s.e.) | Df | F Statistic | *p* Value [5] |
|---|---|---|---|---|---|
| Parity | Primiparous | 0.01 (77) | 1 | 0.0000002 | 0.999 |
| | Multiparous | 0.01 (77) | | | |
| Time period [1] | Period 1 | 0.001 (14.3) | 1 | 0.0000002 | 0.999 |
| | Period 2 | 0.13 (0.05) | | | |
| Assistance [2] | Assisted | 0.01 (78.5) | 1 | 0.002 | 0.962 |
| | Unassisted | 0.01 (77.2) | | | |
| Behavior [3] | | | 4 | 8.7 | <0.001 |
| Time period × assistance | Period 1/Assisted | 0.001 (13.1) | 1 | 0.04 | 0.850 |
| | Period 1/Unassisted | 0.001 (15.6) | | | |
| | Period 2/Assisted | 0.14 (0.09) | | | |
| | Period 2/Unassisted | 0.13 (0.05) | | | |
| Assistance × behavior | Assisted/Contractions | 0.000001 (0.01) | 4 | 0.04 | 0.998 |
| | Assisted/Lying | 0.41 (0.08) | | | |
| | Assisted/Shuffle | 0.07 (0.04) | | | |
| | Assisted/Standing | 0.43 (0.07) | | | |
| | Assisted/Walking | 0.02 (0.02) | | | |
| | Unassisted/Contractions | 0.000001 (0.02) | | | |
| | Unassisted/Lying | 0.38 (0.08) | | | |
| | Unassisted/Shuffle | 0.08 (0.04) | | | |
| | Unassisted/Standing | 0.40 (0.08) | | | |
| | Unassisted/Walking | 0.03 (0.05) | | | |
| Time period × behavior | Period 1/Contractions | 0 (0) | 4 | 0.5 | 0.715 |
| | Period 1/Lying | 0.48 (0.04) | | | |
| | Period 1/Shuffle | 0.07 (0.02) | | | |
| | Period 1/Standing | 0.40 (0.04) | | | |
| | Period 1/Walking | 0.04 (0.02) | | | |
| | Period 2/Contractions | 0.14 (0.07) | | | |
| | Period 2/Lying | 0.32 (0.09) | | | |
| | Period 2/Shuffle | 0.09 (0.06) | | | |
| | Period 2/Stand | 0.43 (0.10) | | | |
| | Period 2/Walking | 0.02 (0.03) | | | |
| Time period × behavior × assistance [4] | | | 3 | 0.5 | 0.685 |

[1] Period 1 was observations obtained 4 to 6 weeks before lambing when the sheep came indoors for close monitoring, and Period 2 was observations obtained before the ewe gave birth. [2] Births were either assisted or unassisted by farm staff. [3] Predicted mean values shown in Figure 1. [4] Predicted mean values shown in Figure 2. [5] Significance was attributed at *p* < 0.05.

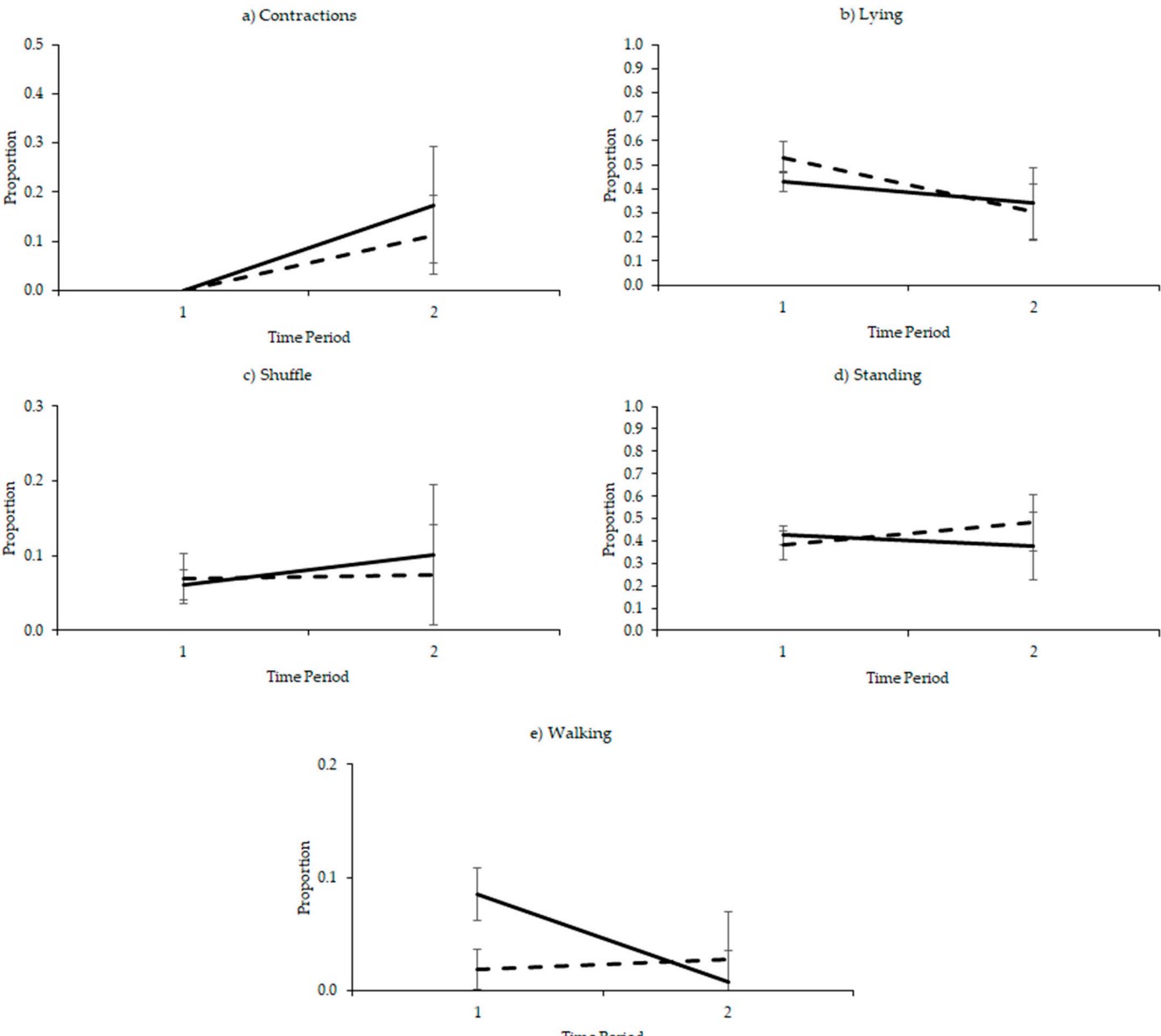

**Figure 2.** Predicted mean (±SEM) proportion of time that were (**a**) contractions, (**b**) lying, (**c**) shuffling, (**d**) standing and (**e**) walking behavior for assisted (dashed line) or non-assisted (solid line) lambing in time Periods 1 or 2, with Period 2 ending with the lambing event.

Differences were also found in the frequency of behaviors ($p < 0.001$) with standing (0.40) and shuffling (0.28) being the most frequent, with other behaviors being 0.09 or less across time periods studied (Figure 3).

In the time period before lambing, the frequency of lying and contraction bouts increased and the standing and walking bouts decreased ($p < 0.001$; Table 2), with a higher frequency of walking bouts in sheep that had an assisted lambing ($p < 0.01$; Figure 4). There was no effect of parity on frequency of behaviors.

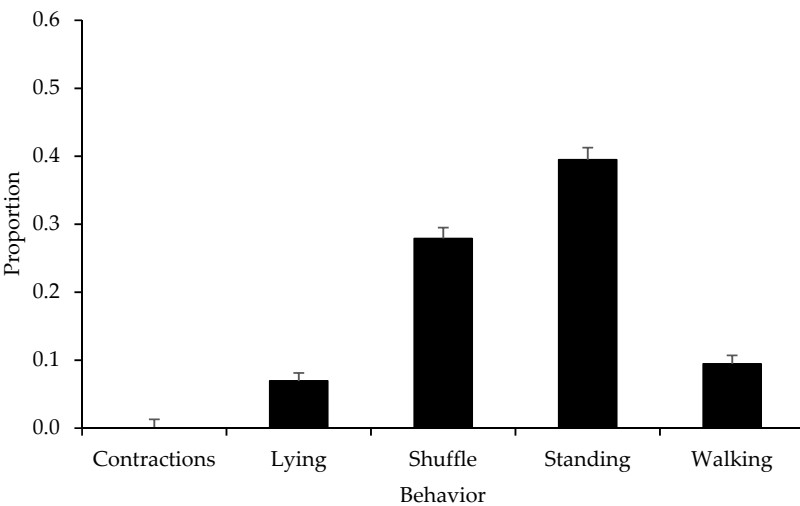

**Figure 3.** Predicted mean (±SEM) proportion of observations for different behaviors.

**Table 2.** Effects of parity, time period, and lambing assistance on the frequency of behaviors as a proportion of observations.

| Variable | | Mean (s.e.) | Df | F Statistic | *p* Value [5] |
|---|---|---|---|---|---|
| Parity | Primiparous | 0.02 (41.4) | 1 | 0.0000001 | 0.999 |
| | Multiparous | 0.02 (41.4) | | | |
| Time period [1] | Period 1 | 0.002 (7.8) | 1 | 0.000001 | 0.999 |
| | Period 2 | 0.17 (0.01) | | | |
| Assistance [2] | Assisted | 0.02 (38.7) | 1 | 1.5 | 0.219 |
| | Unassisted | 0.02 (44.3) | | | |
| Behavior [3] | | | 4 | 53.4 | <0.001 |
| Time period × assistance | Period 1/Assisted | 0.002 (8.3) | 1 | 0.05 | 0.822 |
| | Period 1/Unassisted | 0.002 (7.4) | | | |
| | Period 2/Assisted | 0.18 (0.01) | | | |
| | Period 2/Unassisted | 0.16 (0.02) | | | |
| Assistance × behavior | Assisted/Contractions | 0.000001 (0.01) | 4 | 0.7 | 0.612 |
| | Assisted/Lying | 0.06 (0.02) | | | |
| | Assisted/Shuffle | 0.29 (0.03) | | | |
| | Assisted/Standing | 0.41 (0.03) | | | |
| | Assisted/Walking | 0.12 (0.02) | | | |
| | Unassisted/Contractions | 0.000001 (0.01) | | | |
| | Unassisted/Lying | 0.08 (0.02) | | | |
| | Unassisted/Shuffle | 0.27 (0.02) | | | |
| | Unassisted/Standing | 0.38 (0.02) | | | |
| | Unassisted/Walking | 0.08 (0.02) | | | |
| Time period × behavior | Period 1/Contractions | 0 (0) | 4 | 17.1 | <0.001 |
| | Period 1/Lying | 0.03 (0.01) | | | |
| | Period 1/Shuffle | 0.30 (0.03) | | | |
| | Period 1/Standing | 0.46 (0.03) | | | |
| | Period 1/Walking | 0.20 (0.02) | | | |
| | Period 2/Contractions | 0.19 (0.02) | | | |
| | Period 2/Lying | 0.17 (0.02) | | | |
| | Period 2/Shuffle | 0.26 (0.02) | | | |
| | Period 2/Standing | 0.33 (0.02) | | | |
| | Period 2/Walking | 0.04 (0.01) | | | |
| Time period × behavior × assistance [4] | | | 3 | 5.0 | <0.01 |

[1] Period 1 was observations obtained 4 to 6 weeks before lambing when the sheep came indoors for close monitoring, and Period 2 was observations obtained before the ewe gave birth. [2] Births were either assisted or unassisted by farm staff. [3] Predicted mean values shown in Figure 3. [4] Predicted mean values shown in Figure 4. [5] Significance was attributed at *p* < 0.05.

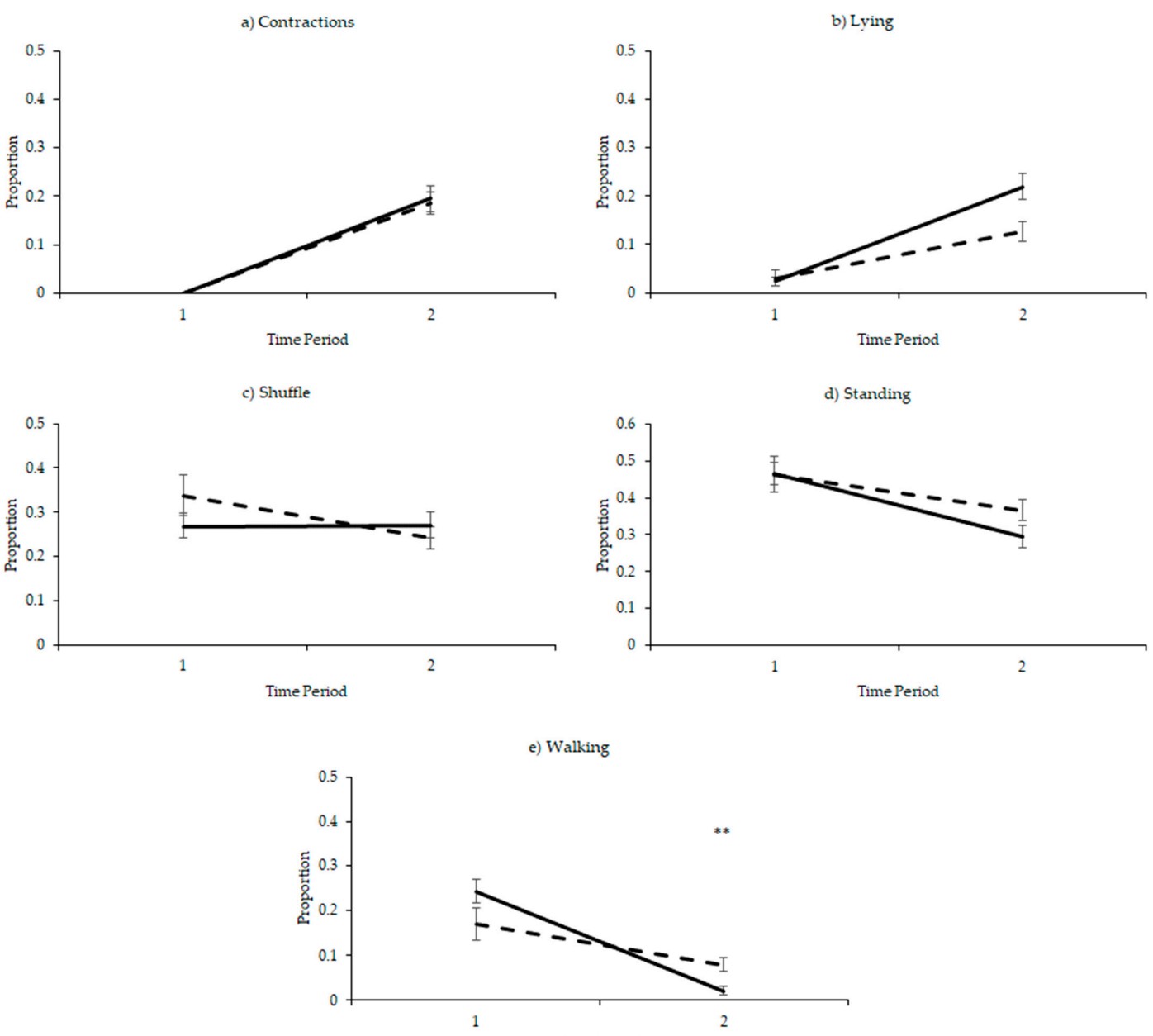

**Figure 4.** Predicted mean (±SEM) proportion of observations that were (**a**) contractions, (**b**) lying, (**c**) shuffling, (**d**) standing and (**e**) walking behavior for assisted (dashed line) or unassisted (solid line) lambing in time Periods 1 or 2, with Period 2 ending with the lambing event. ** $p < 0.01$.

## 4. Discussion

The current study found that sheep spend most of their time either standing or lying during pregnancy. There are surprisingly few studies investigating the duration and frequency of behaviors of pregnant sheep. The current study suggests pregnant ewes spend about 10 h per day lying and 10 h per day standing. This is similar to other ruminant animals such as cattle when indoors, which have been found to spend between 10–12 h per day lying [19]. The duration of behaviors did not appear to change during parturition; however, the frequency of lying bouts, including contractions, increased in the period before lambing. The sheep in the current study were within the last trimester of pregnancy, and the lack of general movement can be expected in heavily pregnant animals with little

distance to their food. Previous studies have shown that there are few differences in the behavior of ewes before and during parturition because of factors such as breed, age of ewe, nutrition, climate or location [16]. The current study also found no significant effect of parity. In cows, Barrier et al. [13] found assisted animals displayed more frequent contractions than those that were unassisted, and that this was often because of awkward positioning of the calf at birth. A general state of restlessness is common in animals as parturition approaches and can also be seen in cow studies, characterized by an increase in lying frequencies as observed in the current study, general increased activity, reduced feed intake, and an intensified stress response [12,14]. In the current study, inclusion of additional behaviors, such as eating and drinking, would have been useful information as these behaviors are known to change as lambing approaches. However, they could not reliably be observed in sheep that were group housed as in the current study. A study by Black and Krawczel [20] found a higher lying frequency was associated with difficult calvings and that the cows that were not exercised were more uncomfortable during parturition. The sheep with an assisted birth in the current study had a higher frequency of walking bouts compared to unassisted births, which suggests restless behavior patterns. Generally, the sheep before lambing in the current study reduced their standing and walking bouts as lying bouts increased. Fogarty et al. [17] found a general increase in walking behavior and frequency of posture changes (i.e., standing and lying) in ewes before lambing. The change in frequencies of standing, lying and walking may provide useful indicators for tracking the progression of birth.

Sheep are often managed in large groups. This makes close inspection of individual animals difficult and timing of observations important for animal husbandry. Therefore, to enhance a farmer's capacity to manage individual animals in large groups, and detect animals who are ill or injured, animal tracking technology has been developed [21,22]. Sensor technologies are not however free from challenges; these devices are extremely sensitive and can be prone to damage from the dirt and dust that comes with farm environments. Their success will rely on the cost-benefit for livestock farming and added value to farm operations such as supporting the intense monitoring of parturition during day or night periods [23]. Sensor technologies could benefit sheep production by allowing more frequent and effective behavior observations at this key stage. Increased behavior monitoring would be extremely beneficial during parturition, as mortality will affect both animal welfare and farm productivity. In sheep, accelerometers have previously been used to detect behavioral states such as high and low general activity or some combinations of lying, standing, grazing, walking and/or running [3,17,24]. Use of an accelerometer, with machine learning, has been found to accurately predict 91% of lambing events within 3 h of birth based on body posture alone [25]. Sensors detecting key behaviors, such as those studied, present new opportunities for a continuous and real-time objective measurement in farm animals [23].

The current study involved a relatively small flock of 17 ewes because of challenges associated with complete video surveillance of animals over two time periods, and obstruction of view in group housing. Although the number of animals studied may have affected the results of this study, they appear consistent with other animal studies as mentioned above. Multiple cameras may have helped increase surveillance coverage and increased the number of animals studied. However, the results from this study suggest that observing changes in lying bouts and detection of contractions could assist farmers in monitoring parturition to enhance sheep husbandry.

## 5. Conclusions

This current study investigating group housed ewes during late pregnancy found an increased frequency of lying bouts, including contractions, before lambing. Pregnant ewes spent a large proportion of their time either lying or standing, with a higher frequency of standing and shuffling bouts. Ewes that required assistance at lambing had more walking

bouts compared to ewes that were unassisted. The monitoring of behavioral patterns, such as lying and contractions, could be used as an alert to the progress of parturition.

**Author Contributions:** Conceptualization, M.J.B. and G.T.; methodology, M.J.B. and G.T.; software, J.M.; validation, J.M.; formal analysis, B.E.W. and M.J.B.; investigation, B.E.W. and M.J.B.; resources, M.J.B.; data curation, M.J.B., K.R.S., Z.J.H. and J.M.; writing—original draft preparation, B.E.W. and M.J.B.; writing—review and editing, B.E.W. and M.J.B.; visualization, J.M.; supervision, M.J.B.; project administration, M.J.B.; funding acquisition, M.J.B. and G.T. All authors have read and agreed to the published version of the manuscript.

**Funding:** This research was funded by the Douglas Bomford Trust, the Engineering and Physical Sciences Research Council and the Biotechnology and Biological Sciences Research Council.

**Institutional Review Board Statement:** The study was conducted according to the guidelines of the Declaration of Helsinki, and approved by the Animal Ethics Committee at the University of Nottingham (approval number 198, 2018).

**Informed Consent Statement:** Not applicable.

**Data Availability Statement:** The analyzed datasets are available from the corresponding author on request.

**Conflicts of Interest:** The authors declare no conflict of interest.

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
