# Peer review of "Changes in Sheep Behavior before Lambing"

_agriculture, doi:10.3390/agriculture11080715_

Round 1

Reviewer 1 Report

ABSTRACT:

Lines 15-16: This study only evaluated the behavior of the animals. Despite this, importante parameters could be assesses such as: ingestive behavior, concentrations of stress-indicating blood metabolites, as well as physiological parameters (heart rate, respiratory rate, rectal temperature, thermographic evaluation. Together, all these results can indicate changes in the behavior of animals, and cal also help to explain more precisely the results obtained in studies of this nature, in different species of ruminants.

Lines: 17-20: before mentioning the animal evaluations, I suggest inserting the description of the animals used (breed, age, weight, standard deviation of weight) in this study, as well as the information on the experimental design and duration of the experimente (adaptation and data collection periods).

Lines 19-20: to carry out this data collection, how were the animals handled? They were housed in collective stalls? It needs to better describe this information, as it directly interferes with the results obtained.

Lines 20-21: before describing the collections, it is important to give a brief idea about the experimental diets offered. Similarly in the abstract section, it was not mentioned in the material and methods.

Line 29: similarly to other scientific journals, probably the keywords must be arranged in alphabetical order

INTRODUCTION

Although references from the last 5 years have been used, the introduction can be better described to highlight the relevance of the study. In lines 52-53, it was mentioned that few studies were conducted in sheep. However, it is important to mention them and highlight how this study contributes to the scientific literature, or the gap that still exists regarding the theme in the sheep species.

Line 54: Before describing the aim of the study, it is important to highlight about the hypothesis. Besides, why were behavioral observations recorded during two separate periods of 4 to 6 weeks before calving and at the calving of 17 ewes? How were these periods stipulated?

MATERIAL AND METHODS

Lines 58-59: The study was carried out with the approval of the ethics committee, but it is essential to include the protocol number, since animals were used in the experiment.

Line 61: before describing the methods of data collection in a visual form item 2.1, important information needs to be inserted in this section of the material and methods such as:

a- climatic conditions (temperature, humidity, altitude, longitude) at the place where this study was carried out? All of this information is important and can have a direct effect on the study results and changes in the behavior of the animals. This was not mentioned in the material and methods.

b- Housing of animals: the animals were confined in a collective pen, but it does not mention anything in relation to the size, presence of drinking fountains and feeding troughs.

c- Duration of the experiment: it was only described that it was conducted from February to March 2019, but it does not describe in detail the period of adaptation or how many days were destined to data collection.

d- management procedures: What procedures were used in the animals throughout the experiment? vaccination, deworming, weighing...

e- Experimental design: It was not clear which design was used and which treatments were evaluated.

f- Experimental diets: What was the diet offered to the animals? Chemical composition, proportion of ingredients, information on requirements (was the NRC used?)

g- Laboratory analyses: What are the laboratory analyzes of the ingredients and diet offered to the animals?

h- Feed intake adjustment: If the animals were managed in collective pens, how was the consumption adjustment made, were indicators used?

i- Line 94: Statistical analysis: what was the level of significance used? Also, triple interaction was analyzed? (Table 1: Time period × behavior × assistance). Why is the information in the material and methods not described?

RESULTS: Lines 113-163

The results were objectively described, but they need to be better explained regarding the information present in the tables and figures, as mentioned below:

Tables 1 and 2: need to describe briefly in the footnote what period 1, period 2, assisted, Unassisted means, in addition to stating the significance level of the results to know whether or not they were significant.

Figures: In general they need to be enlarged, to improve the visualization. In addition, no legend was inserted, and the main statistical values ​​to demonstrate the difference or not between the “assisted” and “unassisted” groups.

DISCUSSION: Lines 165-211

The discussion section needs to be improved to justify the results obtained in this study. An important point to be highlighted is that although camera evaluation was used, the number of experimental units was small and possibly interfered with the results obtained, as described by the authors: Lines 207-211: “The current study involved a relatively small flock of sheep due to challenges associated with complete video surveillance of animals, and obstruction of view”.

CONCLUSION

Lines 213-220: This section needs to be done more objectively, without including information on the results. Furthermore, given what was highlighted by the authors in the discussion, it is important to note that this method of video assessment has limitations when a high number of animals are handled collectively which can interfere with the results obtained.

DECISION: Although current references have been included in the work, the quality of the study and relevance is still compromised, and it is necessary to improve writing in general, especially in terms of material and methods.  It is also important to highlight that although it is not a long research paper, there is a high level of similarity of the current study with others previously conducted.

Author Response

We thank the reviewer for the detailed comments received. All comments have been addressed as shown below, with major changes to the paper.

ABSTRACT:

Lines 15-16: This study only evaluated the behavior of the animals. Despite this, importante parameters could be assesses such as: ingestive behavior, concentrations of stress-indicating blood metabolites, as well as physiological parameters (heart rate, respiratory rate, rectal temperature, thermographic evaluation. Together, all these results can indicate changes in the behavior of animals, and cal also help to explain more precisely the results obtained in studies of this nature, in different species of ruminants.

AU: Yes the study focused on visual observations of behaviour associated with sheep nearing lambing. Further detailed measures could be obtained in a different study, however, this study aimed to look a visual changes in duration and frequency of behaviour for the potential to use vision technology based on findings. “Behavioural observations” has been added to the objective to make this clearer and “visual” monitoring to line 17.

Lines: 17-20: before mentioning the animal evaluations, I suggest inserting the description of the animals used (breed, age, weight, standard deviation of weight) in this study, as well as the information on the experimental design and duration of the experimente (adaptation and data collection periods).

AU: This information is included in the Material and Methods as the abstract is limited to 200 words and there is not enough space to add this extra information. The average bodyweight of the ewes has been added to line 87 as “The average bodyweight of ewes at the start of the study was 58 (s.d. 12) kg.” as requested. The breed and numbers of primiparous and multiparous ewes are included in the Material and Methods.

Lines 19-20: to carry out this data collection, how were the animals handled? They were housed in collective stalls? It needs to better describe this information, as it directly interferes with the results obtained.

AU: This information is included in the Material and Methods as the abstract is limited to 200 words and there is not enough space to add this extra information. The sheep were weighed at the start when being housed as a group, which has been added to line 85 as “When the sheep were housed at the start of the study they were weighed, but after this the sheep were not handled until they had given birth.”

Lines 20-21: before describing the collections, it is important to give a brief idea about the experimental diets offered. Similarly in the abstract section, it was not mentioned in the material and methods.

AU: As mentioned above the abstract is limited to 200 words and lacks space for more detail. Further explanation of the diet fed has been added to the Material and Methods at lines 89 to 95 as “Throughout the study the sheep were group fed ad libitum haylage consisting of 486, 548, 138, 59 g/kg for dry matter, neutral detergent fibre, crude protein and sugar, re-spectively, and 9.7 MJ/kg for metabolisable energy, and 350 g/day a mix of oats with wheat distillers grains consisting of 860, 250, 228, 60 g/kg for dry matter, neutral de-tergent fibre, crude protein and sugar, respectively, and 13.4 MJ/kg for metabolisable energy (Sciantec Analytical Services, Cawood, UK). The diet was about 75% forage.”

Line 29: similarly to other scientific journals, probably the keywords must be arranged in alphabetical order

AU: Reordered as suggested.

INTRODUCTION

Although references from the last 5 years have been used, the introduction can be better described to highlight the relevance of the study. In lines 52-53, it was mentioned that few studies were conducted in sheep. However, it is important to mention them and highlight how this study contributes to the scientific literature, or the gap that still exists regarding the theme in the sheep species.

AU: Further detail has been added in relation to previous studies in sheep and the need for this research as “While there are several studies on changes in cattle behavior before calving [10,12-15], there are few studies on pregnant sheep [16-17]. The need for further research into enhanced monitoring approaches of sheep during parturition has been identified by others [17], and assist a stockperson during this critical period. The hypothesis of the current study was that there is a change in sheep behavior before giving birth, and this change can be visually observed. This insight may assist lambing management and future monitoring technologies.” at lines 53 to 61.

Line 54: Before describing the aim of the study, it is important to highlight about the hypothesis. Besides, why were behavioral observations recorded during two separate periods of 4 to 6 weeks before calving and at the calving of 17 ewes? How were these periods stipulated?

AU: A hypothesis has been added as “The hypothesis of the current study was that there is a change in sheep behavior before giving birth, and this change can be visually observed.” at line 58 as suggested. Regarding the monitoring periods, further explanation and clarity has been added to the Material and Methods section as requested by another reviewer.

MATERIAL AND METHODS

Lines 58-59: The study was carried out with the approval of the ethics committee, but it is essential to include the protocol number, since animals were used in the experiment.

AU: This has been added at line 69.

Line 61: before describing the methods of data collection in a visual form item 2.1, important information needs to be inserted in this section of the material and methods such as:

a- climatic conditions (temperature, humidity, altitude, longitude) at the place where this study was carried out? All of this information is important and can have a direct effect on the study results and changes in the behavior of the animals. This was not mentioned in the material and methods.

b- Housing of animals: the animals were confined in a collective pen, but it does not mention anything in relation to the size, presence of drinking fountains and feeding troughs.

c- Duration of the experiment: it was only described that it was conducted from February to March 2019, but it does not describe in detail the period of adaptation or how many days were destined to data collection.

d- management procedures: What procedures were used in the animals throughout the experiment? vaccination, deworming, weighing...

e- Experimental design: It was not clear which design was used and which treatments were evaluated.

f- Experimental diets: What was the diet offered to the animals? Chemical composition, proportion of ingredients, information on requirements (was the NRC used?)

g- Laboratory analyses: What are the laboratory analyzes of the ingredients and diet offered to the animals?

AU: Section 2.1 has been revised based on comments from reviewers as “A total of 17 pregnant ewes were monitored using video camera surveillance (5 Mp, 30 m IR. Hikvision HD Bullet; Hangzhou, China) at the Nottingham University Farm (Sutton Bonington, Leicestershire, UK; 52.8282°N 1.2485°W, 48m a.s.l) when indoors before lambing from February to March 2019. The ewes monitored were predominantly Lleyn breed, with 9 primiparous and 8 multiparous. Normal husbandry procedure for the flock was followed, whereby all sheep came indoors for closer monitoring as a single group at about 6 weeks before lambing, and returned to pasture after lambing. The sheep were group housed on straw bedding. A single camera was used to obtain continuous video footage of each ewe. The camera position allowed full coverage of the area and at an approximate 45-degree angle looking into the sheep pen. When the sheep were housed at the start of the study they were weighed, but after this the sheep were not handled until they had given birth. The average bodyweight of ewes at the start of the study was 58 (s.d. 12) kg. Throughout the study the sheep were group fed ad libitum haylage consisting of 486, 548, 138, 59 g/kg for dry matter, neutral detergent fibre, crude protein and sugar, respectively, and 9.7 MJ/kg for metabolisable energy, and 350 g/day a mix of oats with wheat distillers grains consisting of 860, 250, 228, 60 g/kg for dry matter, neutral detergent fibre, crude protein and sugar, respectively, and 13.4 MJ/kg for metabolisable energy (Sciantec Analytical Services, Cawood, UK). The diet was about 75% forage. Need for a birth to be assisted by farm staff was recorded for each ewe. The average daily temperature was 5.7 oC, rainfall was 2.9 mm and humidity was 90% during the study period.”

h- Feed intake adjustment: If the animals were managed in collective pens, how was the consumption adjustment made, were indicators used?

AU: The sheep were fed as a group the same diet throughout with ad libitum forage supplement with a cereal mix throughout as added to Material and Methods lines 89 to 94. No alteration in feeding was made.

i- Line 94: Statistical analysis: what was the level of significance used? Also, triple interaction was analyzed? (Table 1: Time period × behavior × assistance). Why is the information in the material and methods not described?

AU: Significance level has been added at the end of the Material and Methods. All three fixed effects are now discussed in the material and methods and included in equation 1. Equation 1 gives the fitted model to the analysis.

RESULTS: Lines 113-163

The results were objectively described, but they need to be better explained regarding the information present in the tables and figures, as mentioned below:

Tables 1 and 2: need to describe briefly in the footnote what period 1, period 2, assisted, Unassisted means, in addition to stating the significance level of the results to know whether or not they were significant.

AU: Significance level has been added as suggested. Footnotes to tables 1 and 2 have been added as suggested to explain time periods and assistance.

Figures: In general they need to be enlarged, to improve the visualization. In addition, no legend was inserted, and the main statistical values ​​to demonstrate the difference or not between the “assisted” and “unassisted” groups.

AU: Figures have been enlarged as suggested. The labels for the figures are in the figure title, for example “Figure 2. Predicted mean (± SEM) proportion of time that were a) contractions, b) lying, c) shuffle, d) standing and e) walking behavior for assisted (dashed line) or non-assisted (solid line) lambing in time periods 1 to 2, with period 2 ending with the lambing event” and the significance is given in the results text and table 1 and 2.

DISCUSSION: Lines 165-211

The discussion section needs to be improved to justify the results obtained in this study. An important point to be highlighted is that although camera evaluation was used, the number of experimental units was small and possibly interfered with the results obtained, as described by the authors: Lines 207-211: “The current study involved a relatively small flock of sheep due to challenges associated with complete video surveillance of animals, and obstruction of view”.

AU: This paragraph at line 251 to 258 has been altered to reflect the comments received as “The current study involved a relatively small flock of 17 ewes due to challenges associated with complete video surveillance of animals over two time periods, and obstruction of view in group housing. While the number of animals studied may have affected the results of the study, the results appear consistent with other animal studies as mentioned above. Multiple cameras may have helped increase surveillance coverage and increased the number of animals studied. However, the results from this study suggest that observing changes in lying bouts and detection of contractions could assist farmers in monitoring parturition to enhance sheep husbandry. Inclusion of additional behaviors, such as eating and drinking, could also be useful information if they can be reliably observed.” Further discussion has been added.

CONCLUSION

Lines 213-220: This section needs to be done more objectively, without including information on the results. Furthermore, given what was highlighted by the authors in the discussion, it is important to note that this method of video assessment has limitations when a high number of animals are handled collectively which can interfere with the results obtained.

AU: The conclusions has been revised based on comments received as “This current study investigating group housed ewes in late pregnancy found an increased frequency of lying bouts, including contractions, before lambing. Pregnant ewes spent a large proportion of their time either lying or standing, with a higher frequency of standing and shuffle bouts. Ewes that required assistance at lambing had more walking bouts compared to ewes that were unassisted. The monitoring of behavioral patterns, such as lying and contractions, could be used as an alert to the progress of parturition.”

DECISION: Although current references have been included in the work, the quality of the study and relevance is still compromised, and it is necessary to improve writing in general, especially in terms of material and methods.  It is also important to highlight that although it is not a long research paper, there is a high level of similarity of the current study with others previously conducted.

AU: The paper has had major revision to the detail of the study, which should improve clarity and comparison with other studies.

Reviewer 2 Report

This manuscript is to assess the duration and frequency of behaviors of pregnant ewes as they approached lambing. It is important to understand behavioral changes prior to birth for enhanced monitoring at this critical stage in the animal’s life. The experimetal design of this study is simple and clear. I only have one suggestion, is it possible to include eating behaviors and drinking? Or, at least discuss such behaviors in introduction or future prospects.

Author Response

We thank the reviewer for their comment which has been addressed as shown below.

This manuscript is to assess the duration and frequency of behaviors of pregnant ewes as they approached lambing. It is important to understand behavioral changes prior to birth for enhanced monitoring at this critical stage in the animal’s life. The experimetal design of this study is simple and clear. I only have one suggestion, is it possible to include eating behaviors and drinking? Or, at least discuss such behaviors in introduction or future prospects.

AU: A sentence has been added to the discussion as “Inclusion of additional behaviors, such as eating and drinking, would have been useful information in the current study, however, they could not reliably be observed in sheep that were group housed in the current study.” Including eating and drinking in the sheep would have added to the study but it is very difficult to observe eating and drinking from the video footage used in the sheep housed in a group with ad libitum feed. The aim was to keep the environment normal.

Reviewer 3 Report

agriculture-1283744-peer-review-v1 “Changes in sheep behavior prior to lambing” Beatrice E. Waters, John McDonagh, Georgios Tzimiropoulos, Kimberley R. Slinger, Zoë J. Huggett and Matt J. Bell

The aim of this study was to assess the duration and frequency of behaviours of pregnant ewes as they approached lambing. This is a worthwhile objective as there is limited information of behaviors of sheep during the third trimester.

My comments are below.

I am concerned about the use of the 4-6 weeks before lambing as period 2. Wouldn’t this be best as period 1? Using 4-6 weeks is really determining what the “normal” behaviours would be and then comparing this to “at lambing” behaviours. It is not very clear the reasons for the two periods.

Title: replace “prior to” with “before”

Abstract:

Not very well written. Conclusions require more elaboration – was this for period 1 or 2 or combined.

L16 replace “prior to” with “before”

L18 replace “prior to” with “before”

L22-24 “The study” at start and end of sentence – reword. This reports on behaviour overall which is not clear to the reader (Fig 1 and Table 1).

L24 replace “prior to” with “before”

Introduction:

L54-55 try not to use one sentence paragraphs. Expand this to include something about methodology.

Materials & methods

This is where I get confused.

L 59 “before the study commenced.”

L64 replace “prior to” with “before”

L68 replace “prior to” with “before”

L73-84 Suggest period 1 is 4-6 weeks and period 2 is lambing.

L84 “Seven behaviours” but only list 5?

L96 should be 170 not 160.

L100 “link function was fitted”

L110-111 try not to use one sentence paragraphs.

Results

Figure 1 caption

L123 “time spent in”

L144 “being the most frequent”. Delete “studied”

L150 replace “prior to” with “before”

Discussion

L167-168 there are quite a few studies on sheep behviours but not in pregnant sheep. Suggest rewording this sentence.

L170 “indoors, that have”

L173 replace “prior to” with “before”

L176 replace “due to” with “because of”

L179 replace “due to” with “because of”

L207 replace “due to” with “because of”

Conclusion

Suggest using some of this in Abstract.

L214 “This current study” and replace “prior to” with “before”

Author Response

We thank the reviewer for their comments, which has lead to major changes to the paper. All comments have been addressed as shown below.

The aim of this study was to assess the duration and frequency of behaviours of pregnant ewes as they approached lambing. This is a worthwhile objective as there is limited information of behaviors of sheep during the third trimester.

My recommendation is that this manuscript requires major revision. My comments are below.

I am concerned about the use of the 4-6 weeks before lambing as period 2. Wouldn’t this be best as period 1? Using 4-6 weeks is really determining what the “normal” behaviours would be and then comparing this to “at lambing” behaviours. It is not very clear the reasons for the two periods.

AU: We have swapped period 1 and 2 to be in time order as suggested. This has been altered throughout the paper to reflect the change. Further explanation of the two-time period as been added as suggested by reviewer 1. The sheep normally come indoors as a group about 6 weeks prior to lambing (depending on actual lambing date) for closer monitoring.  

Title: replace “prior to” with “before”

Abstract:

Not very well written. Conclusions require more elaboration – was this for period 1 or 2 or combined.

AU: The abstract has been revised to reflect comments as “The aim of this study was to assess the duration and frequency of behavioral observations of pregnant ewes as they approached lambing. An understanding of behavioral changes before birth may provide opportunities for enhanced visual monitoring at this critical stage in the animal’s life. Behavioral observations for 17 ewes in late pregnancy were recorded during two separate time periods, which were 4 to 6 weeks before lambing, and prior to giving birth. The sheep were continuously monitored under 24-hour video surveillance. The behaviors of standing, lying, walking, shuffle and contractions were recorded for each animal during both time periods. A generalized linear mixed model was used to assess differences in duration and frequency of behaviors. Over both time periods, the ewes spent a large proportion of their time either lying (0.40) or standing (0.42), with a higher frequency of standing (0.40) and shuffle (0.28) bouts than other behaviors. In the time period before giving birth, the frequency of lying and contraction bouts increased and the standing and walking bouts decreased, with a higher frequency of walking bouts in ewes that had an assisted lambing. The monitoring of behavioral patterns, such as lying and contractions, could be used as an alert to the progress of parturition.”

L16 replace “prior to” with “before”

AU: Replaced as suggested.

L18 replace “prior to” with “before”

AU: Replaced as suggested.

L22-24 “The study” at start and end of sentence – reword. This reports on behaviour overall which is not clear to the reader (Fig 1 and Table 1).

AU: Reworded as suggested.

L24 replace “prior to” with “before”

AU: Replaced as suggested.

Introduction:

L54-55 try not to use one sentence paragraphs. Expand this to include something about methodology.

AU: An additional sentence has been added as “The sheep studied followed normal husbandry procedure of being housed as a group at about 6 weeks before lambing to allow for closer monitoring.”

Materials & methods

This is where I get confused.

L 59 “before the study commenced.”

AU: Revised as suggested.

L64 replace “prior to” with “before”

AU: Section has been revised.

L68 replace “prior to” with “before”

AU: Section has been revised.

L73-84 Suggest period 1 is 4-6 weeks and period 2 is lambing.

AU: Changed as suggested.

L84 “Seven behaviours” but only list 5?

AU: Corrected.

L96 should be 170 not 160.

AU: Corrected.

L100 “link function was fitted”

AU: Edited.

L110-111 try not to use one sentence paragraphs.

AU: Significance has been added as “The back-transformed predicted means for behavior duration and frequency were expressed as the proportion of total time or count during each time period. Significance was attributed at P<0.05.”

Results

Figure 1 caption

L123 “time spent in”

AU: Edited.

L144 “being the most frequent”. Delete “studied”

AU: Edited as suggested as “Differences were also found in the frequency of behaviors (P<0.001) with standing (0.40) and shuffle (0.28) being the most frequent, with other behaviors being 0.09 or less across time periods studied.”

L150 replace “prior to” with “before”

AU: Changed as suggested.

Discussion

L167-168 there are quite a few studies on sheep behviours but not in pregnant sheep. Suggest rewording this sentence.

AU: Added ‘pregnant’ to the sentence as “There are surprisingly few studies investigating the duration and frequency of behaviors of pregnant sheep”.

L170 “indoors, that have”

AU: Changed.

L173 replace “prior to” with “before”

AU: Changed as suggested.

L176 replace “due to” with “because of”

AU: Changed as suggested.

L179 replace “due to” with “because of”

AU: Changed as suggested.

L207 replace “due to” with “because of”

AU: Changed as suggested.

Conclusion

Suggest using some of this in Abstract.

AU: This is already included in the abstract.

L214 “This current study” and replace “prior to” with “before”

AU: Changed as suggested.

Round 2

Reviewer 1 Report

As suggested, the authors made several corrections throughout the manuscript and in the work's conclusions. Despite this, other corrections need to be done to be improved the quality and understanding of the article, mainly in the material and methods as described below:

MATERIAL AND METHODS

  • The authors included the protocol number. However, it is necessary to mention the year when the ethics committee was approves, as usually seen in other scientific articles. Housing of animals: the animals were confined in a collective pen, but it is also important to mention if the collective pens were equipped with drinking fountains and feeding troughs, and how many.
  • Description animals: authors included information about the animals used, however, it is necessary to describe the average year of animals in each group (primiparous and multiparous) and the weight of each group.
  • How animals were assigned to treatments? What was the statistical design used in this study? It important to mention these information in the material and methods.
  • Management procedures: It is also necessary to mention about other management procedures, such as identification, vaccination, deworming of the animals.
  • Experimental diets: Although it was corrected the information about the diets, is necessary to describe better the analyses used to evaluate the chemical composition, and the information about the formulation of diets regarding the requirements (which NRC was used?). Because the quality of the diet affects the behavior of animals, it is very important that the composition be presented in a detailed and complete manner.
  • Feed intake adjustment: Authors made some corrections in this information. However, it is necessary to describe better how the consumption was adjusted. In this study animals were housed in collective pens, so it is necessary to use to use a marker such as titanium dioxide, chromium oxide, among others, to estimate intake.

RESULTS:

  • Tables 1 and 2: It is necessary to include the P-value in the footnotes of tables. It was not described.
  • Figures: Authors made corrections in the figures, which were enlarged. Despite this, it is necessary to include information about SEM, and P-values in order to see the differences between assisted (dashed line) or non-assisted groups.

DECISION: Although the authors made most part of the corrections suggested it is also necessary to include more information to improve the quality of the manuscript.

Author Response

We thank the reviewer for further comments and amendments, which are each addressed below. 

As suggested, the authors made several corrections throughout the manuscript and in the work's conclusions. Despite this, other corrections need to be done to be improved the quality and understanding of the article, mainly in the material and methods as described below:

MATERIAL AND METHODS

The authors included the protocol number. However, it is necessary to mention the year when the ethics committee was approves, as usually seen in other scientific articles. Housing of animals: the animals were confined in a collective pen, but it is also important to mention if the collective pens were equipped with drinking fountains and feeding troughs, and how many.

AU: Year of approval has been added, and access to water and feed troughs as requested.

Description animals: authors included information about the animals used, however, it is necessary to describe the average year of animals in each group (primiparous and multiparous) and the weight of each group.

AU: The average age and weight for primiparous and multiparous ewes has been added as “When the sheep were housed at the start of the study they were weighed, marked with a number for camera observations, and vaccinated for pasteurella and clostridial disease, but after this the sheep were not handled until they had given birth. The average age of primiparous ewes was 1.9 (s.d. 0.03) years and multiparous 4.3 (s.d. 1.0) years. The average bodyweight of primiparous ewes was 57.9 (s.d. 2.6) kg and multiparous 64.2 (s.d. 6.4) kg.”

How animals were assigned to treatments? What was the statistical design used in this study? It important to mention these information in the material and methods.

AU: There was no specific design or treatments applied. The sheep were housed as a single group as a kept as a small flock for student experience. The sheep followed normal farm procedures to ensure normal behaviour was observed. The approached used is described in the Material and Methods and statistical analysis with fixed effects studied.  

Management procedures: It is also necessary to mention about other management procedures, such as identification, vaccination, deworming of the animals.

AU: Further detail has been added as suggested “When the sheep were housed at the start of the study they were weighed, marked with a number for camera observations, and vaccinated for pasteurella and clostridial disease, but after this the sheep were not handled until they had given birth.”

Experimental diets: Although it was corrected the information about the diets, is necessary to describe better the analyses used to evaluate the chemical composition, and the information about the formulation of diets regarding the requirements (which NRC was used?). Because the quality of the diet affects the behavior of animals, it is very important that the composition be presented in a detailed and complete manner.

AU: The chemical analysis of the diet was done by a commercial lab (Sciantec Analytical) as mentioned in the Material and Methods. Further detail on estimation of feed requirements has been added as requested. Intake was estimated as mentioned below and in the Material and Methods.

Feed intake adjustment: Authors made some corrections in this information. However, it is necessary to describe better how the consumption was adjusted. In this study animals were housed in collective pens, so it is necessary to use to use a marker such as titanium dioxide, chromium oxide, among others, to estimate intake.

AU: Feed intake was not measured but based on an estimate. The sheep had ad lib access to forage and the intake of forage was not measured. This could be useful information in a future study. As mentioned above, the animals were not handled during the observation period and additional measures such as this would require faecal sampling to determine intakes. The same diet was fed throughout to minimise disruption – further explanation has been added as “The feed intake requirement was estimated as 2% of bodyweight (about 1.2 kg/day) with the diet providing 11 MJ/kg ME and 160 g/kg crude protein.”

RESULTS:

Tables 1 and 2: It is necessary to include the P-value in the footnotes of tables. It was not described.

AU: The significance level has been added to table footnotes as suggested “Significance was attributed at P<0.05.”

Figures: Authors made corrections in the figures, which were enlarged. Despite this, it is necessary to include information about SEM, and P-values in order to see the differences between assisted (dashed line) or non-assisted groups.

AU: Apologise, figure 4 was accidently copied as figure 2. Correct figure presented and significance added. The standard errors are shown on all figures with the associated mean values.

DECISION: Although the authors made most part of the corrections suggested it is also necessary to include more information to improve the quality of the manuscript.

AU: Further changes and revisions to the discussion have been made to improve the quality of the paper.

Reviewer 3 Report

The authors have improved this manuscript. See below:

Abstract

L20-22 change to “Standing, lying, walking, shuffle and contraction behaviors were recorded for each animal during both time periods.”

L22-23 delete sentence “A generalized … behaviors.”

Introduction

Okay

Materials & methods

L92-93 “procedures for the flock were”

L100-105 sentence too long. Suggest “Sheep were group fed ad libitum haylage consisting of 9.7 MJ/kg metabolisable energy (ME), 486, 548, 138 and 59 g/kg for dry matter, neutral detergent fibre, crude protein and sugar, respectively (Sciantec Analytical Services, Cawood, UK). Also, sheep were supplemented with 350 g/day oats and wheat distillers grain mix (13.4 MJ/kg ME, 860, 250, 228, and 60 g/kg neutral detergent fibre, crude protein, and sugar, respectively).”

L107 delete “period”.

L110-153 “Two observation periods were used to investigate changes in behaviour: Period 1 (4-6 weeks before lambing) and Period 2 (at lambing). There were 10 hours of annotated video recordings for each ewe from Period 1 and three hours before the first lamb was born for Period 2. Tracking of sheep in Period 2 before lambing was challenging because of the similar appearance and behavioral changes.”

Results

Improved

Discussion

L297-300 “In this current study, inclusion of additional behaviors, such as eating and drinking, would have been useful information as these behaviors are known to change as lambing approaches. However, they could not reliably be observed in sheep that were group housed as in the current study.”

L305 -306 “in ewes before lambing.”

L313 “Their success”

L316 delete “of production”

L328 “of this study, they appear”

Conclusions

L358 “ewes during late”

Author Response

We thank the reviewer for the suggested edits to make the paper better. Each point has been addressed below and the paper revised.

The authors have improved this manuscript. See below:

Abstract

L20-22 change to “Standing, lying, walking, shuffle and contraction behaviors were recorded for each animal during both time periods.”

AU: Changed as suggested.

L22-23 delete sentence “A generalized … behaviors.”

AU: Deleted as suggested.

Introduction

Okay

Materials & methods

L92-93 “procedures for the flock were”

AU: Corrected.

L100-105 sentence too long. Suggest “Sheep were group fed ad libitum haylage consisting of 9.7 MJ/kg metabolisable energy (ME), 486, 548, 138 and 59 g/kg for dry matter, neutral detergent fibre, crude protein and sugar, respectively (Sciantec Analytical Services, Cawood, UK). Also, sheep were supplemented with 350 g/day oats and wheat distillers grain mix (13.4 MJ/kg ME, 860, 250, 228, and 60 g/kg neutral detergent fibre, crude protein, and sugar, respectively).”

AU: Changed as suggested.

L107 delete “period”.

AU: Deleted.

L110-153 “Two observation periods were used to investigate changes in behaviour: Period 1 (4-6 weeks before lambing) and Period 2 (at lambing). There were 10 hours of annotated video recordings for each ewe from Period 1 and three hours before the first lamb was born for Period 2. Tracking of sheep in Period 2 before lambing was challenging because of the similar appearance and behavioral changes.”

AU: Revised as suggested.

Results

Improved

Discussion

L297-300 “In this current study, inclusion of additional behaviors, such as eating and drinking, would have been useful information as these behaviors are known to change as lambing approaches. However, they could not reliably be observed in sheep that were group housed as in the current study.”

AU: Revised as suggested.

L305 -306 “in ewes before lambing.”

AU: Changed as suggested.

L313 “Their success”

AU: Changed.

L316 delete “of production”

AU: Deleted.

L328 “of this study, they appear”

AU: Corrected.

Conclusions

L358 “ewes during late”

AU: Changed.

Round 3

Reviewer 1 Report

As suggested, the authors made several corrections throughout the manuscript and in the work's conclusions. Despite this, other corrections need to be done to be improved the quality and understanding of the article,mainly in the material and methods as described below:

MATERIAL AND METHODS

  • How animals were assigned to treatments? What was the statistical design used in this study? It important to mention these information in the material and methods. • Management procedures: As I mentioned before, it is necessary to mention about other management procedures, such as identification and deworming of the animals and treatment against ectoparasites, as is generally described. • Experimental diets: Although it was corrected the information about the diets, is necessary to describe better the analyses used to evaluate the chemical composition, and the information about the formulation of diets regarding the requirements (which NRC was used?). Because the quality of the diet affects the behavior of animals, it is very important that the composition be presented in a detailed and complete manner. • Feed intake adjustment: Although some corrections were previously made in this part of the manuscript it is necessary to describe better how the consumption was adjusted. As I mentioned before, in this study animals were housed in collective pens, so it is important and must be included in the manuscript about the use of a marker such as titanium dioxide, chromium oxide, among others, to estimate intake.   DECISION: Although the authors made most part of the corrections suggested it is also necessary to include more information to improve the quality of the manuscript.

Author Response

We thank the reviewer for their comments, which have each been addressed below and amendments made to the paper.

As suggested, the authors made several corrections throughout the manuscript and in the work's conclusions. Despite this, other corrections need to be done to be improved the quality and understanding of the article, mainly in the material and methods as described below:

MATERIAL AND METHODS

How animals were assigned to treatments? What was the statistical design used in this study? It important to mention these information in the material and methods. 

AU: There were specific treatments for the study but the flock is managed to have similar numbers of older and younger ewes to studies. A sentence has been added at line 71 as “The study was designed to have similar numbers of primiparous and multiparous ewes” to be clearer how the flock is managed for research purposes. The flock is for student studies and experience, and hence a small but manageable size for this purpose. A mixed model approach was used for the statistical analysis as mentioned in section 2.3 with fixed effects of assistance, time period, behavior and parity.

Management procedures: As I mentioned before, it is necessary to mention about other management procedures, such as identification and deworming of the animals and treatment against ectoparasites, as is generally described. 

AU: Additional information has been added to clarify that no individual identification for each animal was recorded and no additional heath treatments were required at lines 79-83 as “When the sheep were housed at the start of the study they were weighed, marked with a number for camera observations and individual identification recorded, and vaccinated for pasteurella and clostridial disease, but after this the sheep were not handled until they had given birth. The sheep did not receive any other heath treatments, such for endo or ectoparasites, during the study.”

Experimental diets: Although it was corrected the information about the diets, is necessary to describe better the analyses used to evaluate the chemical composition, and the information about the formulation of diets regarding the requirements (which NRC was used?). Because the quality of the diet affects the behavior of animals, it is very important that the composition be presented in a detailed and complete manner. 

AU: The analysis was done using near-infrared spectroscopy by a commercial company as stated at lines 88-89 as “(Sciantec Analytical Services, Cawood, UK; using near-infrared spectroscopy analysis).”

Feed intake adjustment: Although some corrections were previously made in this part of the manuscript it is necessary to describe better how the consumption was adjusted. As I mentioned before, in this study animals were housed in collective pens, so it is important and must be included in the manuscript about the use of a marker such as titanium dioxide, chromium oxide, among others, to estimate intake.   

AU: As mentioned before no marker or measurement of feed intake was made in this study. The feed intake was estimated as 2% of bodyweight and fed at a flat-rate through out the study. This section has been reworded to make it clearer “The diet was about 75% forage on a dry matter basis. The feed was allocated as 2% of the average bodyweight for the group of ewes (about 1.2 kg/day), with the diet formulated based on an estimated energy and protein requirement of 11 MJ/kg ME and 160 g/kg crude protein in the diet [18]. The same diet was fed throughout the study and the amount allocated to the group was reduced as sheep gave birth and were removed from the lambing pen.”